# Evaluation of Deep Learning Model Architectures for Point-of-Care Ultrasound Diagnostics

**DOI:** 10.3390/bioengineering11040392

**Published:** 2024-04-18

**Authors:** Sofia I. Hernandez Torres, Austin Ruiz, Lawrence Holland, Ryan Ortiz, Eric J. Snider

**Affiliations:** Organ Support and Automation Technologies Group, U.S. Army Institute of Surgical Research, Joint Base San Antonio, Fort Sam Houston, San Antonio, TX 78234, USA; sofia.i.hernandeztorres.ctr@health.mil (S.I.H.T.); austin.j.ruiz7.ctr@health.mil (A.R.); lawrence.a.holland7.ctr@health.mil (L.H.); ryan.f.ortiz.ctr@health.mil (R.O.)

**Keywords:** ultrasound imaging, artificial intelligence, medical triage, image interpretation, deep learning, medical imaging

## Abstract

Point-of-care ultrasound imaging is a critical tool for patient triage during trauma for diagnosing injuries and prioritizing limited medical evacuation resources. Specifically, an eFAST exam evaluates if there are free fluids in the chest or abdomen but this is only possible if ultrasound scans can be accurately interpreted, a challenge in the pre-hospital setting. In this effort, we evaluated the use of artificial intelligent eFAST image interpretation models. Widely used deep learning model architectures were evaluated as well as Bayesian models optimized for six different diagnostic models: pneumothorax (i) B- or (ii) M-mode, hemothorax (iii) B- or (iv) M-mode, (v) pelvic or bladder abdominal hemorrhage and (vi) right upper quadrant abdominal hemorrhage. Models were trained using images captured in 27 swine. Using a leave-one-subject-out training approach, the MobileNetV2 and DarkNet53 models surpassed 85% accuracy for each M-mode scan site. The different B-mode models performed worse with accuracies between 68% and 74% except for the pelvic hemorrhage model, which only reached 62% accuracy for all model architectures. These results highlight which eFAST scan sites can be easily automated with image interpretation models, while other scan sites, such as the bladder hemorrhage model, will require more robust model development or data augmentation to improve performance. With these additional improvements, the skill threshold for ultrasound-based triage can be reduced, thus expanding its utility in the pre-hospital setting.

## 1. Introduction

During trauma situations, it is common for medical care personnel to utilize triage procedures to characterize casualties or wounded patients to effectively distribute the appropriate level of care for each patient [1]. This is particularly relevant in combat casualty care as mass casualty situations require rapid triage and are anticipated on the future battlefield [2]. This is further complicated by a challenged airspace that will require prolonged field care in the battlefield of up to 72 h as has been common during the Ukraine–Russia conflict [3]. Medical imaging is a commonly used tool for triage as it enables examining internal injuries that may not be apparent. One common medical imaging-based triage application in military medicine is extended-focus assessment with sonography for trauma (eFAST), which can reliably assess free fluid in the abdominal and thoracic cavities [4]. One advantage of the eFAST exam is that it uses ultrasound (US) equipment, which can be portable, has lower power requirement, and is more affordable compared to other medical imaging approaches [5], giving eFAST considerable availability at or near the point of injury [6]. As such, eFAST can enable rapid triage in a pre-hospital or combat casualty care setting.

While eFAST procedures are commonly used, there are several aspects that should be taken into consideration. The primary challenge is that the procedure depends on the availability of US technicians that are needed for proper image acquisition, interpretation and triage decisions. In an extended emergency trauma or mass casualty situation, it is anticipated that there will be shortages of medical personnel that can perform an eFAST scan and make a diagnosis [2]. The longer it takes for a wounded soldier to be medically evaluated and transported to a higher echelon of care or treatment, the greater the logistical burden and strain on resources becomes, thus making it more difficult to effectively triage. While less trained personnel can perform an eFAST exam, the diagnoses are only as good as the images captured, and there is known subjectivity variance in diagnostic accuracy between sonographers [7]. Given the urgency of identifying trauma in an efficient manner, this can lead to diagnostic bias when making triage decisions.

With the implementation of artificial intelligence (AI) models, there could be a significant force multiplier benefit in scaling diagnostic capabilities in military medicine. Having AI models interpret images for injury diagnosis could effectively alleviate the responsibilities from specially trained medical providers to combat medics, thus enabling more personnel to aid with emergency triage. Due to the variability in emergencies on the battlefield, it is difficult to predict surges in trauma emergencies that are consequentially considered as mass casualty incidents; therefore, having a model to provide diagnostic predictions would help with response time in triaging. Having a trained model to classify diagnostics for US images can also objectify the process of making decisions during triage and lower the skill threshold needed to obtain diagnosis predictions once US images are acquired.

The utilization of AI—especially in research and medicine—has accelerated exponentially in recent times, propelled by big data and edge computing technology, paired with the continuous development of improved computer hardware capabilities [8,9]. Given the context of AI’s revolutionary effect in the mainstream, there are currently a variety of AI models that have been developed for diagnostics, medical examination procedures, and treatment personalization [10]. For medical imaging, AI is used to assist in determining or identifying the status of patients across the most common medical imaging techniques, such as computed tomography [11,12], magnetic resonance imaging [13,14,15], and US imaging [16,17,18]. This helps automate a process that would otherwise take more time to complete manually, with a lower margin of subjectivity or bias. As triaging procedures are relatively standardized, it is beneficial to free medical professionals from these tasks to enable them to delegate more time for urgent patients or protocols. This results in a quicker, more effective triaging response in specific, unprecedented circumstances such as mass casualty situations.

In this study, we developed and evaluated AI models for intrabdominal or intrathoracic free fluid detection in US images across some of the standardized eFAST scan locations to automate diagnosis on the battlefield. The main contributions for this study are as follows:A 27-swine subject dataset was curated with positive and negative injury conditions for certain eFAST scan locations.A convolutional neural network (CNN) was optimized for each eFAST scan site using a two-step optimization process, which involved exhaustive and Bayesian optimization of a wide range of hyperparameters.Custom model architectures were compared against lighter models with fewer parameters and larger conventional models using a leave-one-subject-out cross-validation approach.

### Overview of the eFAST Procedure

The eFAST triage methodology is a modification of the original FAST examination, which only assessed intrabdominal hemorrhage and pericardial effusion [19,20]. Four standardized scan points were identified in the FAST methodology for (i) the subxiphoid view to identify pericardial (PC) effusion, (ii) the right upper quadrant (RUQ) view for detection of abdominal hemorrhage in the hepatorenal recess, (iii) the left upper quadrant (LUQ) view for fluid identification in the splenorenal recess, and (iv) the pelvic or bladder (BLD) view for assessing the rectovesical or rectouterine pouch. The extension of this methodology incorporated in the eFAST procedure integrated thoracic US imaging for identifying pneumothorax (PTX) or hemothorax (HTX) injuries as indicated by free air or blood in the pleural space, respectively, diagnosed by US imaging multiple intercostal space views on the left and right chest [20,21].

## 2. Materials and Methods

### 2.1. Data Collection

Research was conducted in compliance with the Animal Welfare Act, the implementing Animal Welfare regulations, and the principles of the Guide for the Care and Use for Laboratory Animals. The Institutional Animal Care and Use Committee at the United State Army Institute of Surgical Research approved all research conducted in this study. The facility where this research was conducted is fully accredited by the AAALAC International. Live animal subjects were maintained under a surgical plane of anesthesia and analgesia throughout the studies. US scans were captured at the eFAST scan points from two approved swine protocols prior to instrumentation, as 10 s brightness (B-) mode clips for all scan sites or motion (M-) mode images over a 5 s window for the thoracic scan regions. The first protocol (*n* = 13 subjects) involved a splenectomy procedure followed by a burn injury to then study the response to burn resuscitation for a 24-h period, after which the subjects were euthanized. The second protocol (*n* = 14 subjects) also started with a splenectomy, followed by two rounds of controlled hemorrhage and fluid resuscitation. After the second resuscitation, US scans were captured for only the thoracic and right upper quadrant (RUQ) scan sites since the subjects were in prone position; after collecting US clips, the subjects were euthanized.

After euthanasia from either protocol, US scans were obtained at every eFAST scan point except for the pericardial site and left upper quadrant (LUQ). For the thoracic region, pneumothorax (PTX) and hemothorax (HTX) injuries were created by inserting air or blood through a triple lumen central venous catheter (Arrow International, Morrisville, NC, USA) placed near the anterior axillary region, connected to tubing and a peristaltic pump. The catheter was introduced using a modified version of the Seldinger technique, by inserting the needle in between the pleural layers. US scans were then collected specifically in the PTX or HTX injury regions as B-mode or M-mode clips. The abdominal hemorrhage injuries were created by reopening part of the abdominal incision and inserting the tubing from the peristaltic pump near the apex of the liver. As fluid pooled near the kidney/liver interface, or around the bladder, US scans were collected in RUQ and BLD regions as positive for injury.

### 2.2. Image Setup

A total of 6 scan sites were evaluated across several AI architectures (described in subsequent sections). The PTX and HTX sites were evaluated for both B-mode (PTX_B, HTX_B) and M-mode (PTX_M, HTX_M) image capture (Figure 1). Images for the LUQ or the cardiac site were not used due to all swine subjects undergoing a splenectomy procedure, and the post-mortem injury model could not replicate these sites.

US scans were collected using either a linear or curvilinear transducer with the Sonosite PX System (Fujifilm, Bothwell, WA, USA), and then organized by protocol, subject, scan region, and injury classification as either positive or negative. All further pre-processing was performed using MATLAB R2023b (MathWorks, Natick, MA, USA), where the B-mode clips were split into frames, cropped to remove the user interface information, and then resized to 512 × 512 pixels. For the M-mode images, the motion capture segment was split into 25 single-second cropped sections using a rolling window [22]. These sections were then resized to 512 × 512 pixels for consistency. The total number of images across the training dataset is summarized in Table 1.

Before setting up any training for the AI models, images were initially split into image datastores for each scan point using MATLAB. The datastores were implemented to automate categorizing image datasets based on different characteristics of the images such as the scan site, diagnosis, and subject. Three different model training approaches were taken in this effort as shown in Figure 2: exhaustive optimization, Bayesian optimization, and cross-validation assessment. Datasets were spliced into different size clusters for each training approach so that subjects did not have any redundancy between data splits that would otherwise cause data leakage. Total image quantity varied between subject and scan site, to standardize amount of training data, subsets of images with injury were split by scan site. Negative images were subsequently loaded into the data splits to match the number of positive images in each.

### 2.3. Exhaustive Optimization for Training Parameters

The starting convolutional neural network (CNN) model architecture for the optimization approach for each scan site was adapted from a published TensorFlow example [23]. Briefly, the model used three 2D CNN layers with 16, 32, and 64 total filters with a filter size of three for each of three layers. Each layer used a rectified linear unit (ReLU) activator followed by a max pooling layer. The output of the model after the third CNN layer was a fully connected layer with an output size of two which utilized a variable activation layer. Parameters chosen for the first optimization round (Figure 2) focused on training hyperparameters rather than the architecture, apart from the final activation layer. The parameters chosen and the values tested are as follows: (i) batch size—16, 64, 128—the number of images fed into the model with each training iteration; (ii) optimizer—RMSprop [24], ADAM [25], SGDM [26,27]—algorithms used to minimize the error of the model loss function [28]; (iii) the learning rate—0.001, 0.0005, 0.0001—the rate at which the model updates its weights, or “learns”; and (iv) final activator—softmax (Equation (1)) or softplus (Equation (2))—a function that evaluates the feature maps from the final convolution layer before they are fed to the classification layer.
(1)$$y_{r}\left(x\right)=\frac{\exp\left(a_{r}\left(x\right)\right)}{\sum_{j=1}^{k}\exp\left(a_{j}\left(x\right)\right)}$$
(2)$$Y=\log\left(1+e^{X}\right)$$

As shown in Figure 2, the first round of AI training consisted of an exhaustive optimization of training parameters, with different combinations of training options. These approaches used 2 data splits, one as training and validation dataset, and the second as blind test images. From each of the B-mode scan point image sets, 2500 images were selected at random from the training split as well as another 500 images for validation. 500 images were then taken at random from the blind test images to be used for testing models after training. Due to the reduced number of images for M-mode scan points, 1250 images were pulled for training, and 250 images for both the validation and testing datasets.

Before training, random image augmentation was implemented in three ways to add variability to the US images. First, *Y*-axis reflection randomly flipped images about the *Y*-axis. Next, random rotation was implemented, enabling up to 36 degrees of rotation in either direction. The final image augmentation was random scaling, scaling the image within a range from 0.90 to 1.10. For training, a MATLAB script was developed to cycle through each unique combination of optimization parameters and train a model before moving on to the next. A total of 54 unique training runs were conducted in this stage to a maximum of 100 epochs. The models were trained with an early stopping function, called “validation patience” using MATLAB. The function’s purpose is to reduce time spent on improving model performance if it is not continuously improving after a user defined period of epochs. This value was set to 5 epochs, meaning that if the validation loss values did not improve after 5 epochs, the model terminated training early.

To select the best performing model from the first phase of optimization, accuracy scores were taken from predictions on the blind test dataset for all 54 trained models. For each model with an accuracy of 0.5 or higher, 0.5 was subtracted from the score. If the accuracy was below 0.5, a score of 0 was assessed for that model. This approach was taken so that all optimized parameters received a score across each of the 54 trained models. These scores were then summed, resulting in an overall score for each unique parameter. For example, a model with parameters of 16 batch size, RMSprop optimizer, learning rate of 0.001, softmax activator, and 0.74 accuracy would add 0.24 to the overall score for each of these parameters. Once these scores were compiled, the highest score for each parameter was then selected as optimal for moving on to the second phase of optimization.

### 2.4. Bayesian Optimization for Algorithm Architecture

After the initial exhaustive optimization, a second phase was set up to further improve model performance (Figure 2) using Bayesian optimization, focused on the model architecture rather than training parameters. Optimization parameters and their ranges of values for this round are as follows: (i) filter size—integers ranging from 2 to 7—the size of the convolution kernel; (ii) the number of layers—integers ranging from 2 to 6—the number of convolution layers in the model; (iii) the number of filters—integers ranging from 2 to 16—the number of output feature maps for the first convolution layer; (iv) the multiplier—real numbers from 1 to 2—the factor by which the number of filters increases with each layer; and (v) dropout, the rate at which feature maps were randomly removed from the model flow [29], to help prevent overfitting. The dropout layer was added to the end of the model for this phase of optimization, before the final activation and classification layers (integers from 1 to 9, then divided by 10 for creating rates between 10% and 90%).

Training was conducted with the same data setup for exhaustive optimization now using MATLAB’s bayesopt function. This function takes a set of parameters and range of values for each parameter and attempts to find the best combination of parameters to maximize a given output function. In this case, the function we are maximizing is the accuracy value given after testing each trained model on the set of blind test images. Similar to exhaustive optimization, validation patience was set to 5 epochs for a maximum of 100 epochs. Training options were set based on the best performing options from round 1 of optimization. The bayesopt function attempted random combinations of the architecture parameters up to a maximum of 100 unique combinations. With each model trained, the function adjusted the parameters within the set constraints to yield a better accuracy score [30]. The three models with the highest accuracy scores were then chosen as the best performing and evaluated through full LOSO training runs.

### 2.5. Overview of LOSO Training

When evaluating a model’s architecture and its potential to be used in a practical setting, testing its performance on blind datasets is a qualitative method for seeing how well it can generalize new data based on the data it was trained on. Deploying a method called leave-one-subject-out (LOSO) in the model’s training provides a way to test an architecture’s ability to generalize by giving a variety of different training datasets as well as test datasets with blind data. For a given scan point, the swine subjects were split into similarly sized clusters where each cluster is referred to as a subject in the LOSO training set up. By creating five LOSO clusters, the algorithms were able to effectively experience different training and testing datasets. This was used to evaluate how different architectures are able to generalize to datasets [31].

For testing model performance, we compared five different CNN-based architectures for each scan site (Figure 2): (i) the original CNN architecture prior to any optimization, termed “simple CNN”; (ii) the top three Bayesian optimized model architectures, termed “optimized CNN” and only one model results are shown based on which had the highest average accuracy; (iii) the ShrapML CNN architecture, which was previously optimized for US image interpretation [32,33]; (iv) the widely used MobileNetV2 [34], which had the highest performance in previous US imaging applications [35]; and (v) DarkNet53 conventional model architecture [36] as an example of a CNN architecture with more depth that has also performed successfully for US image interpretation [35]. Both DarkNet53 and MobileNetV2 used their pre-trained ImageNet parameter weights.

Similar to the training setup for model optimization, an image database categorized by scan point, image mode type, and swine subject number was used to divide the data up evenly into five clusters by subject number for each scan point (RUQ, BLD, HTX_B, HTX_M, PTX_B, and PTX_M). The five clusters represented subjects in a LOSO training setup, wherein data for four subjects were merged for training and validation while the remaining subject was held out across five LOSO splits. 16,000 total images (8000 positive and negative image labels) were randomly selected from the training data for B-mode. With less images for M-mode data types, only 4000 total images (2000 for each class) were used. The images were augmented with up to 10% random rotation, random *y*-axis reflection, and 10% random image re-scaling. An additional 2000 or 500 images were randomly selected for split validation data for B-mode and M-mode, respectively. For blind test data, 2000 or 500 images were randomly selected for each LOSO setup for B-mode and M-mode, respectively. Training was performed with up to 100 epochs, using a similar validation patience of 5 and training parameters as dictated by the optimization results.

### 2.6. LOSO Model Performance Evaluation and Data Analysis

Model performance was measured by comparing predictions on blind holdout data for each LOSO split versus ground-truth labels. Through this comparison test image results were split into True Positive (*TP*), True Negative (*TN*), False Positive (*FP*), and False Negative (*FN*) labels and were used to construct confusion matrices. Using these labels, performance metrics for accuracy, precision, recall (sensitivity), specificity, and *F1-scores* were calculated using established definitions (Equations (3)–(7)). In addition, the area under the receiver operator characteristic (AUROC) curve was quantified for the positive label category for test images in each LOSO split. Performance metrics were calculated using MATLAB. GraphPad Prism 10.1.2 (La Jolla, CA, USA) was used for statistical analysis of performance metrics for optimization and LOSO training as well as data visualization.
(3)Accuracy=(TN+TP)(TP+TN+FP+FN)
(4)Precision=TP(TP+FP)
(5)Recall=TP rate=TP(TP+FN)
(6)Specificity=TN rate=TN(TN+FP)
(7)F1-score=(2×Recall×Precision)(Recall+Precision)

To better illustrate model explainability, gradient-weighted class activation mapping (GradCAM) was used to generate heat map overlays for *TP*, *TN*, *FP*, and *FN* image labels using MATLAB. Five images of each label were randomly generated for each model architecture, training LOSO, and scan site. The generated GradCAM overlays provide “hot spots” that indicate what areas of the image were most critical to the predicted label to determine if the AI model is accurately tracking injury location to better understand performance [37].

## 3. Results

### 3.1. Optimization of a CNN Architecture for eFAST

We first optimized the CNN model across four main hyperparameters: (i) batch size, (ii) optimizer, (iii) the learning rate, and (iv) final layer activator, in an exhaustive optimization (see methods). Starting with batch size, the smaller 16 image batch size was optimal for all but the HTX_B models which favored a 64 image batch size (Figure 3A). The optimizer and the learning rate were more split across the scan sites but only two of the possible options were selected, with SGDM and 0.005 not being preferred by any scan site (Figure 3B,C). For all scan points but BLD, softmax was selected as the final layer activator, as BLD optimized to the softplus layer (Figure 3D). However, the BLD scan site had poor training performance across all the optimization rounds with 0.61 blind test accuracy being the strongest results. The overall selected features for each scan site are summarized in Table 2.

We further optimized CNN model architecture across a wider range of hyperparameters using a Bayesian optimization approach. Five different hyperparameters were tuned through this methodology—(i) filter size, (ii) the number of layers, (iii) the number of filters, (iv) multiplier, and (v) dropout rate after the fully connected layer. Distributions of the 100 Bayesian optimization runs are shown in Figure 4 for each hyperparameter, to highlight what features per scan site the optimization method focused on. For most scan sites, the top performing optimized architectures had similar setups, with each of the PTX_B models having three CNN layers or each HTX_M model having the same filter size, dropout rate, and node size. For others, the model parameters varied significantly, such as BLD with a wide range selected across all optimized parameters. However, the blind test accuracy was low for optimization once again for BLD, with the top performing model achieving 0.57 accuracy. As a result of heterogeneity in some of the selected parameters for scan sites, we evaluated the top three model setups further using LOSO cross-validation training. A summary of the top three performing model architectures is shown in Appendix A.

### 3.2. Evaluation of Different Model Architectures across eFAST Scan Points

#### 3.2.1. Right Upper Quadrant (RUQ) Models for Abdominal Hemorrhage Injury

For each scan site, seven different model architectures were compared using five LOSO training cuts, resulting in five different blind test results, which were averaged for each scan site and model. Starting with the RUQ scan site, ShrapML had the highest *TP* rate or recall (0.79, Figure 5C), while MobileNetV2 had the highest *TN* rate or specificity at 0.77 (Figure 5D). For assessing feature identification in US images, we use GradCAM overlays to produce images for each confusion matrix category. The simple CNN model was focusing on small features that often traced the boundary of the kidney in *TP* or *TN* predictions (Figure 5A), while ShrapML was often tracking features in the image larger than where hemorrhage would be identified (Figure 5C). The optimized, MobileNetV2, and DarkNet53 models were focused on smaller regions of the image that were often near where abdominal hemorrhage would be identified (Figure 5).

Based on overall accuracy, MobileNetV2 had the strongest blind test performance at 0.79 ± 0.15 but all models except for DarkNet53 were above 0.70 (Figure 5F), different to validation accuracies where the lowest accuracy was for the optimized model at 0.90 ± 0.08 and MobileNetV2, DarkNet53, and ShrapML all having scores of 0.98 or higher (Figure 5F). The blind test performance metrics for each RUQ model, including the three optimized models, are summarized in Appendix A.

#### 3.2.2. Bladder (BLD) Models for Abdominal Hemorrhage Injury

For the BLD scan site, performance remained low for all architectures, but results differed in higher recall for the optimized CNN (0.67, Figure 6B), while the rest of the architectures had higher specificity, with ShrapML having a rate of 0.70 (Figure 6C). Looking at the heat map prediction overlays, the simple CNN model continued to segment out the edges of the region of interest when making accurate predictions (Figure 6A). All other models were tracking the proper BLD region of interest or peripheral region around the bladder for image predictions.

However, the overall blind test accuracy was low for all models, with ShrapML having the highest accuracy at 0.62 ± 0.14, while the simple CNN had the lowest accuracy at 0.52 ± 0.09 (Figure 6F). Based on validation accuracy, the results showed a strong overfitting trend with accuracies above 0.95 for all models except the optimized model with 0.59 ± 0.06 accuracy (Figure 6F). The blind test performance metrics for each BLD model, including the three optimized model setups, are summarized in Appendix A.

#### 3.2.3. Pneumothorax (PTX) Models for Thoracic Scan Site Injury

For PTX injury models, B-mode and M-mode image modalities were separately evaluated to determine if one approach was more readily automated. PTX_B models showed strong recall, evident for the simple CNN, optimized CNN, and MobileNetV2 models, with each reaching above 0.79 for this metric (Figure 7). Conversely, only ShrapML was able to surpass 0.70 specificity scores, with most other models being close to 0.50. On average, the highest blind test accuracy was 0.68, with several model architectures nearing this score threshold (Figure 7F). DarkNet53, MobileNetV2, and ShrapML had higher overfitting tendencies with split validation accuracies at 0.89, 0.92, and 0.98, respectively.

GradCAM overlays were similar for the simple and optimized CNN models with both tracking small features throughout the US images, while ShrapML, MobileNetV2, and DarkNet53 were tracking larger regions (Figure 7). MobileNetV2 most often identified the pleural space, where injury is evident for PTX, but DarkNet53 and ShrapML frequently identified additional regions in the images (Figure 7). The blind test performance metrics for all trained PTX_B models are summarized in Appendix A.

For PTX_M, all models performed well at identifying positive injuries in test image sets, with DarkNet53 reaching 0.96 recall (Figure 8E). MobileNetV2 had the strongest specificity at 0.90 followed closely by DarkNet53 at 0.86, while all other models had lower performance (Figure 8). Evaluating heat map overlays showed a similar trend for the simple and optimized CNN models, where small features are evident in each GradCAM image. The other models identified the pleural line, or the regions below it in the image, where PTX injuries are most easily identified due to lack of lung motion (Figure 8).

Overall, blind test accuracies for MobileNetV2 and DarkNet53 were very strong at 0.89 ± 0.04 and 0.91 ± 0.05, respectively. These models far outperformed the others, with the next highest accuracy at 0.73 ± 0.10 for the optimized model. For split validation accuracies, ShrapML had the highest discrepancy from blind test scores (split validation = 0.97 ± 0.02 vs. blind test = 0.64 ± 0.15) likely indicating high overfitting to training data. The performance metrics for each trained PTX_M model are summarized in Appendix A.

#### 3.2.4. Hemothorax (HTX) Models for Thoracic Scan Site Injury

For HTX B-mode models, performances were skewed toward higher recall and lower specificity. The MobileNetV2 and DarkNet53 models had the highest recall scores at 0.89 and 0.87, respectively (Figure 9D,E), while the optimized model achieved only 0.66 specificity. GradCAM overlays for all models but simple CNN were accurately making decisions based on the pleural space between the rib spacing as would be the standard clinical diagnostic approach (Figure 9A–E). For overall blind test accuracy, HTX B-mode models performed better than PTX B-mode models, with three model architectures surpassing 0.70 accuracy, while no PTX_B models exceeded this performance (Figure 9F). The highest performing model was MobileNetV2 at 0.74 ± 0.10. However, there was still a large overfitting trend when looking at split validation accuracy metrics with all models except the simple CNN (0.88 ± 0.05) model surpassing 0.97 split accuracy scores (Figure 9F). The blind test performance metrics for each HTX_B model, including the three optimized model setups, are summarized in Appendix A.

For HTX_M, all models continued to trend toward higher recall, with DarkNet53 and MobileNetV2 both surpassing 0.90. Specificity trended lower, with a lowest score of 0.63 for ShrapML and the highest value reaching 0.80 for MobileNetV2 (Figure 10A–E). Based on GradCAM analysis, simple and optimized CNN models were tracking lines across the M-mode image vs. larger regions of interest. However, for *TP* results, these lines often coincided with fluid in the pleural space (Figure 10A,B). The other three model architectures were tracking the proper image regions including tracking the expansion of the pleural space due to the presence of fluid (Figure 10C–E).

While HTX_B models outperformed PTX_B, HTX_M models performed worse in comparison to PTX_M, with highest blind test accuracies by MobileNetV2 and DarkNet53 only reaching 0.85 ± 0.07 and 0.81 ± 0.08, respectively (Figure 10F). All other models had blind test accuracies between 0.70 and 0.74. Split validation accuracy trended higher again, with all models surpassing scores of 0.83 (Figure 10F). The blind test performance metrics for all trained HTX_M models are summarized in Appendix A.

## 4. Discussion

Ultrasound imaging is a critical tool in emergency medicine for initial assessment of injuries and triaging patient status for prioritizing medical evacuation resources. The utility for ultrasound imaging can be extended if the skill threshold can be lowered for acquisition and interpretation of scan results so that imaging-based triage can be more common in the pre-hospital military or civilian setting. The eFAST triage application we focus on here is critical for detecting free fluid in the thoracic or abdominal cavities and positive eFAST diagnosis can often require urgent surgical intervention. AI image interpretation models for identifying positive and negatives status at each scan point can streamline this process without needing ultrasonography expertise if high accuracy AI models can be trained for this application. In this effort, we highlighted different approaches and model architectures for each eFAST scan point to optimize their performance.

Overall, AI models trained on a number of the eFAST scan points resulted in high blind test performance, while other scan points proved more challenging in creating a generalized automated solution. Five different model architectures were evaluated using a LOSO cross-validation methodology, and each had mixed performance across the eFAST scan points and imaging modes. The simple network architecture represented a basic CNN model, which may be sufficient for minimal interpretation of image information. The optimized models had their hyperparameters tuned for each scan point application. ShrapML was previously optimized for an ultrasound imaging application and outperformed conventional architectures for shrapnel identification in tissue [33]. MobileNetV2 was selected as it had performed best for shrapnel detection compared to other models [35]. DarkNet53 was an example of a much deeper neural network with many more trainable parameters to evaluate if this design was more suitable for this application.

For the M-mode imaging modality used in the PTX_M and HTX_M scan sites, MobileNetV2 and DarkNet53 outperformed the other architectures. This is an interesting finding as this modality generally enables straightforward diagnosis compared to the B-mode, which would be expected to require simpler model architectures. This result had less to do with the other models performing worse and pertained to MobileNetV2 and DarkNet53 training much better for these scan points, compared to the B-mode equivalent. This is logical in the context that M-mode images are reconstructed from hundreds of B-mode vertical slices, giving the AI model contextual awareness, while B-mode interpretations must rely on a single image frame [38]. This is especially challenging for thoracic scan sites as lung motion can result in the PTX or HTX injury being hidden in a single B-mode frame, which is less likely to be missed in an M-mode image. This temporal image complexity may not be suitable for more simple CNN architectures as suggested by the larger, more complex model architectures outperforming in this work. Furthermore, it could suggest value in having a time series ultrasound image input for improving image training as M-mode results outperformed all B-mode results. Other studies have utilized higher-order inputs to CNN models to include more contextual detail [39,40] or successfully integrate recurrent neural network architecture for better tracking time series details in ultrasound images [41,42].

For the other scan sites, RUQ had a similar performance across most of the trained models, with accuracy in the low 0.70 range. However, split validation accuracy was 0.90 or higher for all architectures, indicating that the models were overfitting the training data and unable to interpret different subject variability in the test data. Of note, the performance across the LOSOs was highly variable for this scan site, with the range of accuracies for the five LOSO runs from 0.87 to as low as 0.60 for MobileNetV2. The training and test noise were potentially inconsistent across data splits, suggesting that strategies to better account for subject variability will be needed. Improved data augmentation can be one strategy to improve these results as it has been successful in improving model performance for various medical applications [43].

The BLD scan point resulted in the worst training performance, with the best model accuracy reaching only 0.62 accuracy. There were a number of data capture and physiological challenges with this scan site that may partially explain this poor performance. First, images captured in live animals occurred after a urinary catheter was placed, which sometimes resulted in small, uniform bladder sizes. While more images were captured post-mortem for this scan site in which the bladder was replenished to a wider range of bladder sizes, the live animal images were only negative for hemorrhage while post-mortem were both positive and negative for abdominal hemorrhage. As a result, the negative BLD live animal images may be creating a very challenging training problem. Another challenge with this scan point is that unlike other scan points, there are two changing variables—bladder size and abdominal hemorrhage severity—compared to the other scan points having a single variable, free fluid quantity. This additional variable makes training models for this scan point challenging and may require additional data augmentation strategies or more robust model architectures to improve performance.

Two of the most important aspects of a model’s training dataset are its size and variability. For future improvements to all model performances, the plan is to tackle the dataset’s issues with both concepts in mind. The dataset size issue is conceptually simple to fix by collecting more images as well as training models on more images. However, larger datasets require a higher computational training burden, making larger, more complex model architectures cumbersome to evaluate. Instead, focus can be set on artificially increasing dataset variability to combat model overfitting. Image augmentations and transformations are the simplest ways of achieving this, without needing to collect more ultrasound images. For ultrasound image classification tasks, oftentimes training data are collected from ultrasound video clips, where very little variability is evident on a frame-by-frame basis. The image transformation performed in this paper was chosen with the idea of producing a level of variability that one may expect from ultrasound images. Rotations and scaling were kept relatively restricted as intuition would lead one to believe that scaling an ultrasound image to one-tenth of its original size would not occur in a practical setting. However, it is possible to see better performance when using data augmentation that would produce a non-useful image when viewed by a human observer [44]. Future data augmentation pursuits include increasing transformation types used while also using optimization techniques to tune their parameters.

## 5. Conclusions

In conclusion, ultrasound medical imaging can be a critical pre-hospital triage tool given its portability and small form factor if tools can be developed to simplify image acquisition and interpretation. The AI models developed in these efforts are steps toward automating image interpretation for an eFAST scan point for detecting free fluid in the thoracicx and abdominal cavity. Next steps toward this effort will improve training strategies to reduce model overfitting and create more generalized diagnostic models. In addition, steps will be taken to translate these models into a real-time format for pairing with handheld ultrasound technology so that predictions can be provided easily to the end-user. This will enable pairing with techniques that are in development to improve image acquisition such as virtual reality or robotic solutions [45,46,47]. Combining the AI models developed in this research effort with image acquisition solutions will drastically lower the skill threshold for ultrasound-based medical triage, enabling its use more readily in pre-hospital and combat casualty care.

## Figures and Tables

**Figure 1 bioengineering-11-00392-f001:**
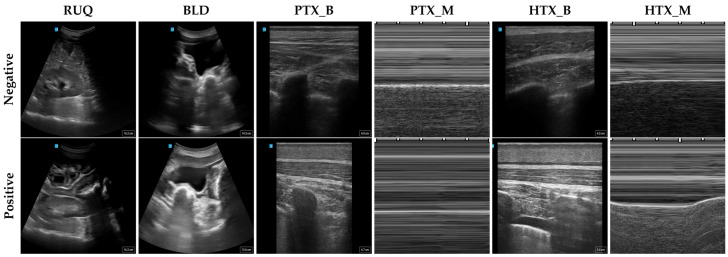
Representative US images for each scan point. Each column shows US images at the different scan points for negative (**top**) or positive (**bottom**) injury state. Please note that the manufacturer logo is present in the top left while the image depth is shown in bottom right corner for B-mode images (RUQ, BLD, PTX_B, and HTX_B). For M-mode images (PTX_M and HTX_M), distance between each tick mark corresponds to 0.2 s.

**Figure 2 bioengineering-11-00392-f002:**
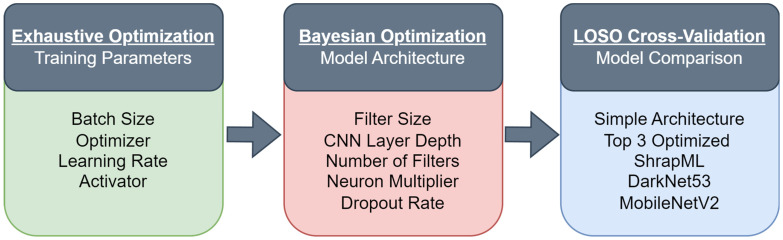
Flowchart of architecture optimization pipeline. Sequence of optimization rounds with the parameters that were varied.

**Figure 3 bioengineering-11-00392-f003:**
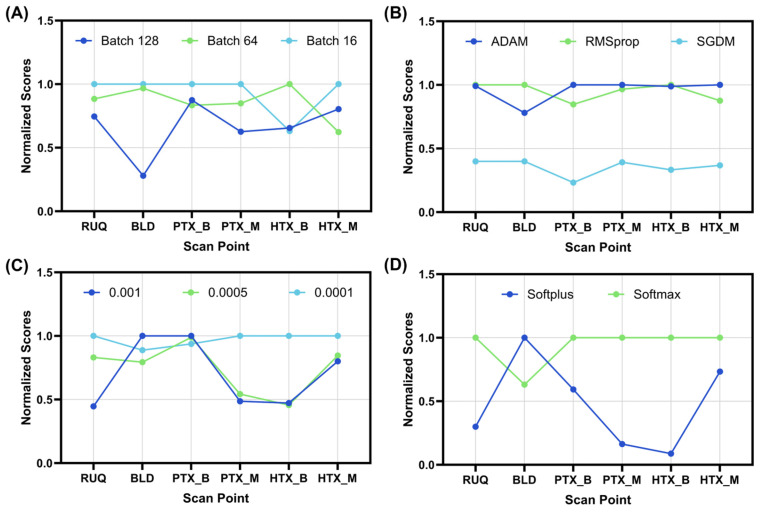
Normalized performance score for the exhaustive optimization for each scan site. Graphical representation of the results for: (**A**) batch size, (**B**) optimizer, (**C**) learning rate, and (**D**) activation function. Results are normalized to the maximum performing model for each scan site, resulting in the data being gated between 0 and 1 (*n* = 54 exhaustive optimization model runs).

**Figure 4 bioengineering-11-00392-f004:**
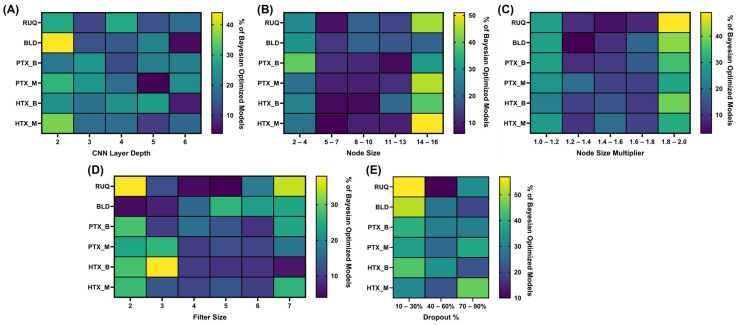
Distribution of values across Bayesian optimization for each scan point and hyperparameter. Results are shown as heatmaps for the frequency of Bayesian iterations (*n* = 100) in which values fell in set bin sizes for (**A**) CNN layer depth, (**B**) node size, (**C**) node size multiplier, (**D**) filter size, and (**E**) dropout percentage.

**Figure 5 bioengineering-11-00392-f005:**
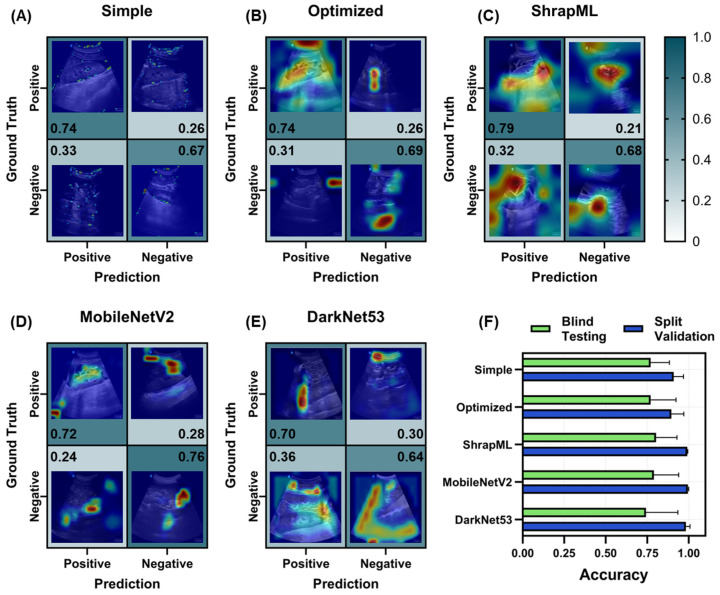
Prediction results from the LOSO training regimen for the RUQ scan site for different AI architectures. (**A**–**E**) Confusion matrices with GradCAM showing AI prediction results for *TP*, *TN*, *FP* and *FN* for (**A**) simple CNN architecture, (**B**) the top optimized CNN model, (**C**) ShrapML, (**D**) MobileNetV2, and (**E**) DarkNet53. Confusion matrix values are shown as the relative amounts to each ground truth category, resulting in the *TP* and *FP* rates shown in the first column and the *FN* and *TN* rates shown in the second column for blind test image predictions. The GradCAM overlay highlights relevant regions for AI prediction in red/yellow tones. (**F**) Summary of accuracy metric scores for blind test and split validation datasets for each model. Results are shown as the mean values across LOSO cross-validation runs (*n* = 5 splits). Error bars denote the standard deviation.

**Figure 6 bioengineering-11-00392-f006:**
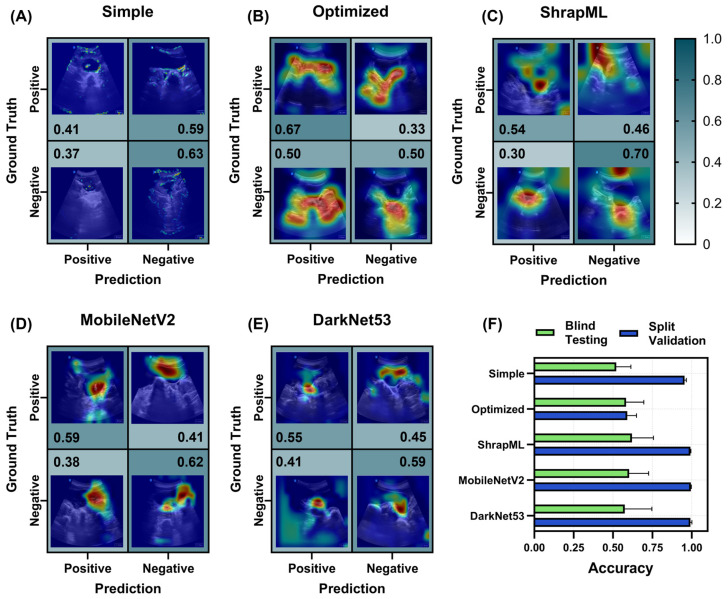
Prediction results from the LOSO training regimen for the BLD scan site for different AI architectures. (**A**–**E**) Confusion matrices with GradCAM showing AI prediction results for *TP*, *TN*, *FP* and *FN* for (**A**) simple CNN architecture, (**B**) the top optimized CNN model, (**C**) ShrapML, (**D**) MobileNetV2, and (**E**) DarkNet53. Confusion matrix values are shown as the relative amounts to each ground truth category, resulting in the *TP* and *FP* rates shown in the first column and the *FN* and *TN* rates shown in the second column for blind test image predictions. The GradCAM overlay highlights relevant regions for AI prediction in red/yellow tones. (**F**) Summary of accuracy metric scores for blind test and split validation datasets for each model. Results are shown as the mean values across LOSO cross-validation runs (*n* = 5 splits). Error bars denote the standard deviation.

**Figure 7 bioengineering-11-00392-f007:**
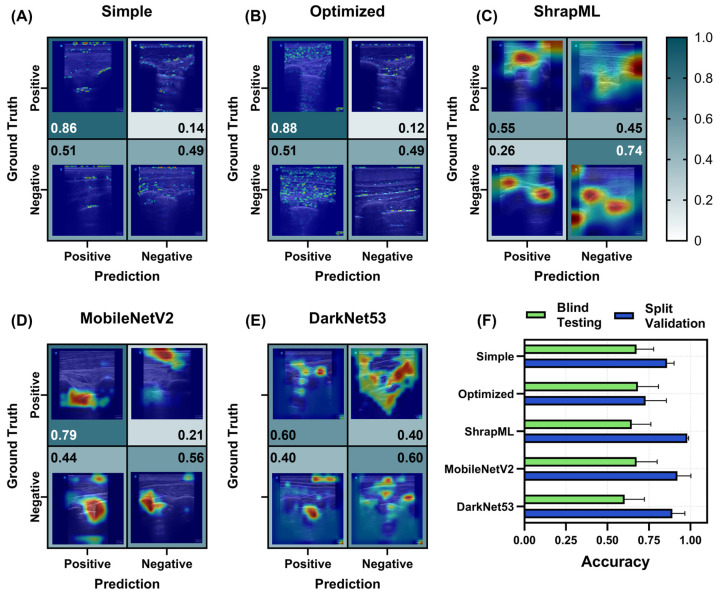
Prediction results from the LOSO training regimen for the PTX_B scan site for different AI architectures. (**A**–**E**) Confusion matrices with GradCAM showing AI prediction results for *TP*, *TN*, *FP* and *FN* for (**A**) simple CNN architecture, (**B**) the top optimized CNN model, (**C**) ShrapML, (**D**) MobileNetV2, and (**E**) DarkNet53. Confusion matrix values are shown as the relative amounts to each ground truth category, resulting in the *TP* and *FP* rates shown in the first column and the *FN* and *TN* rates shown in the second column for blind test image predictions. The GradCAM overlay highlights relevant regions for AI prediction in red/yellow tones. (**F**) Summary of accuracy scores for blind test and split validation datasets for each model. Results are shown as the mean values across LOSO cross-validation training runs (*n* = 5 splits). Error bars denote the standard deviation.

**Figure 8 bioengineering-11-00392-f008:**
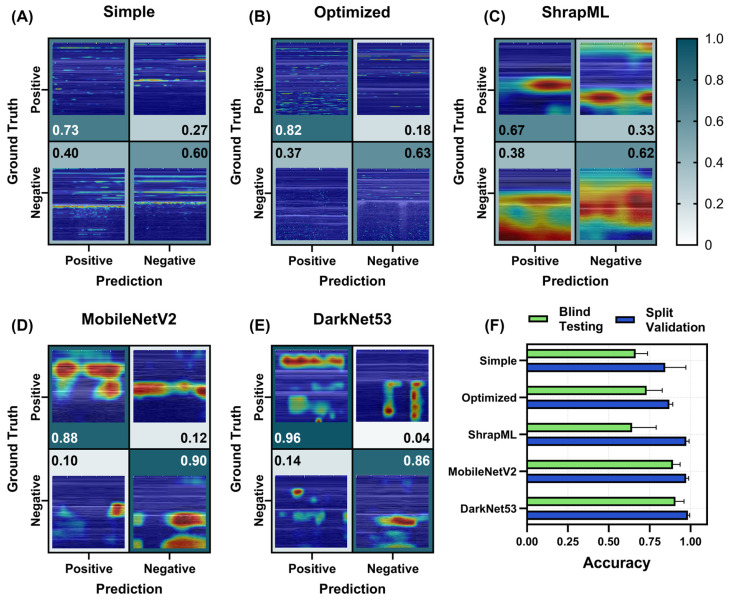
Prediction results from the LOSO training regimen for the PTX_M scan site for different AI architectures. (**A**–**E**) Confusion matrices with GradCAM showing AI prediction results for *TP*, *TN*, *FP* and *FN* for (**A**) simple CNN architecture, (**B**) the top optimized CNN model, (**C**) ShrapML, (**D**) MobileNetV2, and (**E**) DarkNet53. Confusion matrix values are shown as the relative amounts to each ground truth category, resulting in the *TP* and *FP* rates shown in the first column and the *FN* and *TN* rates shown in the second column for blind test image predictions. The GradCAM overlay highlights relevant regions for AI prediction in red/yellow tones. (**F**) Summary of accuracy metric scores for blind test and split validation datasets for each model. Results are shown as the mean values across LOSO cross-validation training runs (*n* = 5 splits). Error bars denote the standard deviation.

**Figure 9 bioengineering-11-00392-f009:**
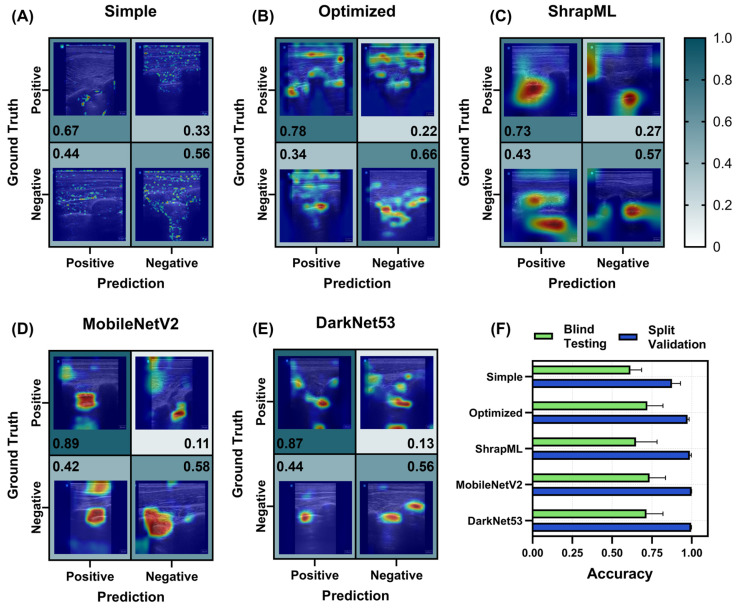
Prediction results from the LOSO training regimen for the HTX_B scan site for different AI architectures. (**A**–**E**) Confusion matrices with GradCAM showing AI prediction results for *TP*, *TN*, *FP* and *FN* for (**A**) simple CNN architecture, (**B**) the top optimized CNN model, (**C**) ShrapML, (**D**) MobileNetV2, and (**E**) DarkNet53. Confusion matrix values are shown as the relative amounts to each ground truth category, resulting in the *TP* and *FP* rates shown in the first column and the *FN* and *TN* rates shown in the second column for blind test image predictions. The GradCAM overlay highlights relevant regions for AI prediction in red/yellow tones. (**F**) Summary of accuracy metric scores for blind test and split validation datasets for each model. Results are shown as the mean values across LOSO cross-validation training runs (*n* = 5 splits). Error bars denote the standard deviation.

**Figure 10 bioengineering-11-00392-f010:**
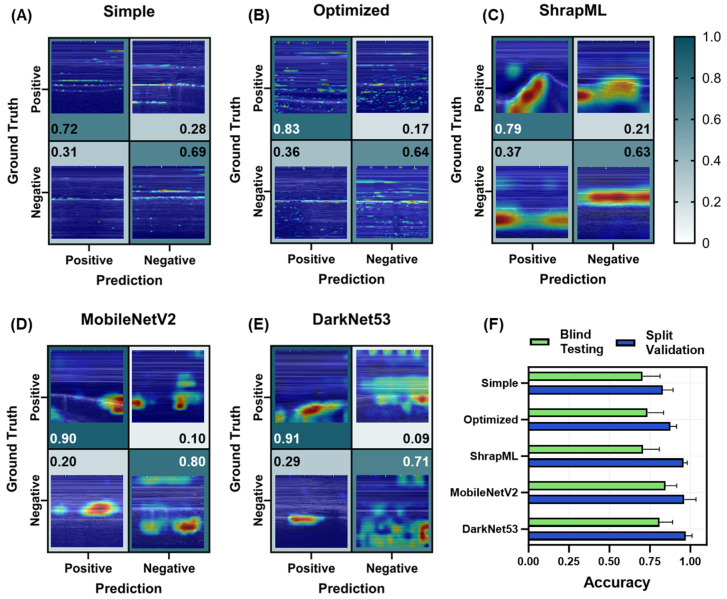
Prediction results from the LOSO training regimen for the HTX_M scan site for different AI architectures. (**A**–**E**) Confusion matrices with GradCAM showing AI prediction results for *TP*, *TN*, *FP* and *FN* for (**A**) simple CNN architecture, (**B**) the top optimized CNN model, (**C**) ShrapML, (**D**) MobileNetV2, and (**E**) DarkNet53. Confusion matrix values are shown as the relative amounts to each ground truth category, resulting in the *TP* and *FP* rates shown in the first column and the *FN* and *TN* rates shown in the second column for blind test image predictions. The GradCAM overlay highlights relevant regions for AI prediction in red/yellow tones. (**F**) Summary of accuracy metric scores for blind test and split validation datasets for each model. Results are shown as the mean values across LOSO cross-validation training runs (*n* = 5 splits). Error bars denote the standard deviation.

**Table 1 bioengineering-11-00392-t001:** Summary of the total number of images for each diagnostic classification and the number of swine subjects for each scan point.

Scan Point	RUQ	BLD	PTX_B	PTX_M	HTX_B	HTX_M
Positive Images	30,000	20,845	34,957	4525	76,431	9368
Negative Images	31,396	22,049	54,420	6425	54,420	6425
Total Number of Images	61,396	42,894	89,377	10,950	130,851	15,793
Subjects	25	21	22	20	25	25

**Table 2 bioengineering-11-00392-t002:** Summary of selected parameters for each eFAST scan point based on exhaustive optimization of the CNN model.

	Batch Size	Optimizer	Learning Rate	Activator
RUQ	16	RMSprop	0.0001	softmax
BLD	16	RMSprop	0.001	softplus
PTX_B	16	ADAM	0.001	softmax
PTX_M	16	ADAM	0.0001	softmax
HTX_B	64	RMSprop	0.0001	softmax
HTX_M	16	ADAM	0.0001	softmax

## Data Availability

The datasets presented in this article are not readily available because they have been collected and maintained in a government-controlled database that is located at the US Army Institute of Surgical Research. As such, this data can be made available through the development of a Cooperative Research & Development Agreement (CRADA) with the corresponding author. Requests to access the datasets should be directed to Eric Snider, eric.j.snider3.civ@health.mil.

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
