# Peer review of "Evaluation of Deep Learning Model Architectures for Point-of-Care Ultrasound Diagnostics"

_bioengineering, 2024, doi:10.3390/bioengineering11040392_

Round 1

Reviewer 1 Report

Comments and Suggestions for Authors

Summary:

The authors proposed to use a deep learning-based eFAST exam system to evaluate free fluids in the chest or abdomen using ultrasound images. The effectiveness of the models varied across different scanning locations.

Comments:

1.      Given the study's relatively small sample size, 3-fold or 5-fold cross-validation methods are highly recommended. This approach would allow for the replication of results.

2.      It is not clear whether the Convolutional Neural Network (CNN) was developed from scratch or if it utilized pre-existing pretrained models. Incorporating transfer learning could potentially enhance the model's performance by starting with pretrained weights from widely available sources such as ImageNet or RadImageNet.

3.      Considering the model outputs probabilities, accuracy may not be the most appropriate metric for evaluation in this scenario. It's suggested that the authors adopt ROC AUC as a more suitable metric, given that accuracy assessments were based on AUC after determining a probability threshold.

Author Response

  1. Given the study's relatively small sample size, 3-fold or 5-fold cross-validation methods are highly recommended. This approach would allow for the replication of results.

We appreciate the review of our submitted manuscript. For cross-validation methods, we used two different approaches. For the initial development of the optimized AI models, we split the subjects into two folds for these optimization studies to accelerate the repetition training of models. However, these models and all other models when fully trained and tested in this study underwent a 5-cross-validation setup where subjects were kept blind across these cuts. This resulted in 5 models developed and average results are shown throughout. We hope this information alleviates these concerns for the reviewer.

  1. It is not clear whether the Convolutional Neural Network (CNN) was developed from scratch or if it utilized pre-existing pretrained models. Incorporating transfer learning could potentially enhance the model's performance by starting with pretrained weights from widely available sources such as ImageNet or RadImageNet.

Some of the models were developed from scratch in this effort and do not have ImageNet weights. Those were the simple architecture, optimized models, and ShrapML. The MobileNetV2 and DarkNet53 models started with their ImageNet pre-trained weights. This information has been added on line 262 in the methods section.

  1. Considering the model outputs probabilities, accuracy may not be the most appropriate metric for evaluation in this scenario. It's suggested that the authors adopt ROC AUC as a more suitable metric, given that accuracy assessments were based on AUC after determining a probability threshold.

ROC AUC was measured and used as a metric in this study and can be found in supplementary tables for each eFAST scan site. Overall, accuracy and AUC results trended in the same order of performance across our trained models. Because the trends were the same, we highlighted accuracy in the manuscript figures for simpler messaging to the reader.

Reviewer 2 Report

Comments and Suggestions for Authors

This work compared several machine learning models for ultrasound image analysis needed in point-of-care diagnosis. The work is scientifically sound and has good application potential. The data analysis is complete, the experimental methodology is clear, and the results are described accurately. Though there are a few issues such as the number of subjects being small and the limited accuracy of the blind test of machine learning models, the issues are due to the limited experimental cost, and the ML algorithm, which are beyond the scope of this paper. So these imperfections are acceptable. Considering the scientific value and application prospect of the research, it is recommended to accept this paper.

Author Response

We appreciate the review of our manuscript.

Reviewer 3 Report

Comments and Suggestions for Authors

The manuscript is technically sound and well-written. Some minor isses were detected.

The main findings is not included in the abstract as well as the comparative improvement compared with literature.

The authors should include the list of the main contributions of this study in Section 1.

Proposed methods should include formal mathematical foundations in detail.

Author Response

  1. The main findings is not included in the abstract as well as the comparative improvement compared with literature.

Thanks for reviewing our manuscript. We modified the abstract wording to better capture the main findings of the study which were that these initial model setups were sometimes sufficient for eFAST scan point whereas other scan points such as the bladder still require modifications to reach improved performance. We hope the modified phrasing addresses this point.

  1. The authors should include the list of the main contributions of this study in Section 1.

A section was added at the end of the introduction to highlight what the main aspects to the research study were at line 84.

  1. Proposed methods should include formal mathematical foundations in detail.

To address this comment, we have added mathematical formulas for the activation layers compared in this study during the model tuning phase, as well as references to the mathematical foundation for the training optimizer functions. We have also added equations to better describe how performance metrics were calculated. All of these changes can be found in the methods section of the manuscript.

Round 2

Reviewer 1 Report

Comments and Suggestions for Authors

the authors address my comments